# Factors Associated with Fear of Falling in Individuals with Different Types of Mild Cognitive Impairment

**DOI:** 10.3390/brainsci12080990

**Published:** 2022-07-26

**Authors:** Pei-Hao Chen, Ya-Yuan Yang, Ying-Yi Liao, Shih-Jung Cheng, Pei-Ning Wang, Fang-Yu Cheng

**Affiliations:** 1Department of Neurology, MacKay Memorial Hospital, Taipei 104, Taiwan; a7662888@gmail.com (P.-H.C.); csjneuro@gmail.com (S.-J.C.); 2Department of Medicine, MacKay Medical College, New Taipei City 252, Taiwan; 3Graduate Institute of Mechanical and Electrical Engineering, National Taipei University of Technology, Taipei 106, Taiwan; 4Institute of Long-Term Care, MacKay Medical College, New Taipei City 252, Taiwan; nina40137@gmail.com; 5Kaifeng Minquan Day Care Center, Taipei 104, Taiwan; 6Department of Gerontological Health Care, National Taipei University of Nursing and Health Sciences, Taipei 112, Taiwan; ianliao1209@gmail.com; 7Department of Physical Therapy and Assistive Technology, National Yang Ming Chiao Tung University, Taipei 112, Taiwan; 8Department of Neurology, Neurological Institute, Taipei Veterans General Hospital, Taipei 112, Taiwan; linda2860@gmail.com; 9Department of Neurology, School of Medicine, National Yang Ming Chiao Tung University, Taipei 112, Taiwan; 10Brain Research Center, National Yang Ming Chiao Tung University, Taipei 112, Taiwan

**Keywords:** mild cognitive impairment, fear of falling, Alzheimer disease, Parkinson disease

## Abstract

Mild cognitive impairment (MCI) is considered an intermediate state between normal aging and early dementia. Fear of falling (FOF) could be considered a risk indicator for falls and quality of life in individuals with MCI. Our objective was to explore factors associated with FOF in those with MCI due to Alzheimer’s disease (AD-MCI) and mild cognitive impairment in Parkinson’s disease (PD-MCI). Seventy-one participants were separated into two groups, AD-MCI (*n* = 37) and PD-MCI (*n* = 34), based on the disease diagnosis. FOF was assessed using the Activities-specific Balance Confidence scale. The neuropsychological assessment and gait assessment were also measured. FOF was significantly correlated with global cognitive function, attention and working memory, executive function, Tinetti assessment scale scores, gait speed, and stride length in the AD-MCI group. Moreover, attention and working memory were the most important factors contributing to FOF. In the PD-MCI group, FOF was significantly correlated with gait speed, and time up and go subtask performance. Furthermore, turn-to-walk was the most important factor contributing to FOF. We noted that FOF in different types of MCI was determined by different factors. Therapies that aim to lower FOF in AD-MCI and PD-MCI populations may address attention and working memory and turn-to-walk, respectively.

## 1. Introduction

Falls among community-dwelling older adults are common events in daily life, and they can lead to disability, hospitalization and even death [1]. Older adults, especially those at risk of falls, may also exhibit peculiar psychological features, such as fear of falls, linked to the experience of negative emotions related to falls [2]. FOF is defined as cautious concern with falling that leads to an individual losing confidence and avoiding activities associated with daily life [3]. Having a FOF was an independent risk factor for falling among persons older than 65 years of age [3]. Therefore, it is important to investigate the factors associated with FOF to incorporate these factors in prevention and rehabilitation programs. Several factors may contribute to FOF among aging adults. Age, sex, history of falls, balance and gait performance and depression are significantly associated with FOF [4]. The completion time in the timed up and go task was also shown to be significantly correlated with FOF [4].

Cognitive impairment has been identified as a risk factor for falls in aging [5]. Mild cognitive impairment (MCI) is considered a clinical stage between the expected cognitive decline in normal aging and the more serious decline in early dementia [6]. Falls are more prevalent in older adults with MCI than in age-matched healthy subjects [7]. A previous study noted that older adults with MCI reported FOF more often than patients with mild Alzheimer’s disease and older adults with healthy cognition [2]. FOF not only leads to an increased risk for falling but also causes restriction and avoidance of activities that eventually result in degenerated physical and mental status [8]. Identifying factors related to FOF in MCI patients may be useful in developing multicomponent strategies to decrease FOF and improve quality of life. However, there is no study investigating the factors associated with FOF in older adults with MCI.

There are many types of MCI. Clinical presentations have shown that they can be amnestic, and they can involve a single nonmemory domain or involve multiple cognitive domains [9]. Each of these clinical presentations may have multiple etiologies, such as degenerative, vascular, metabolic, traumatic, and psychiatric etiologies [9]. Based on etiopathology, the most common neurodegenerative disorders associated with MCI are Alzheimer’s disease and Parkinson’s disease. Patients who are diagnosed with mild cognitive impairment due to AD (AD-MCI) usually have amnestic MCI and positive biomarkers for both Aβ and neuronal injury [10]. On the other hand, based on the guidelines proposed by the International Parkinson and Movement Disorders Society, the definition of mild cognitive impairment in PD (PD-MCI) refers to patients with a diagnosis of Parkinson’s disease, whose cognitive abilities decline and are not caused by other comorbidities or diseases [11]. These two types of MCI have different clinical manifestations because of the different underlying diseases. For example, memory is the most commonly impacted domain among patients with AD-MCI [10]; in contrast, both nonamnestic and amnestic domains of cognition can be affected in persons with PD-MCI [12]. The factors that relate to FOF in these two types of MCI populations may be different. However, there is limited knowledge regarding the contributing factors. Hence, this study aimed to determine factors associated with FOF among people with different types of MCI (i.e., AD-MCI and PD-MCI).

## 2. Materials and Methods

### 2.1. Participants and Study Design

This was an observational, cross-sectional design, and quantitative study. We recruited 71 older adults with MCI from the neurological outpatient clinics of a medical center in Northern Taiwan. We included older adults (age 60 years and older) who were able to walk 10 m independently and to meet the inclusion criteria of MCI (subjective cognitive complaints, a global clinical dementia rating of 0.5, and a clinical dementia rating sum of boxes of 0.5–4.0 [13]). We excluded individuals with dementia, positive psychiatric history or unstable medical conditions and those taking any medications causing cognitive complaints during the past 3 months. A total of 100 participants provided written informed consent prior to enrollment. The study procedures were approved by the ethics committee of the institution (number: 18MMHIS005e). The inclusion process for this study is depicted in Figure 1 (*n* = 71). The participants’ age, sex, education, body mass index, history of fall (have fallen in the last year), and history of metabolic disease were obtained from patient interviews and medical charts. We administered the Barthel Index to explore participants’ self-care activities, for example, transferring, bathing, and toileting [14], and the Instrumental Activities of Daily Living scale to assess participants’ ability to perform task such as cooking, using a telephone, laundry, and handling finances [15].

### 2.2. Defining the MCI Etiologies

The diagnosis for the MCI etiology of each participant was relied on reviewing clinical chart, brain imaging data, neuropsychological tests, and biochemical results. Diagnosis was made for AD-MCI according to the National Institute on Aging and the Alzheimer’s Association workgroup consensus criteria [10]. AD-MCI was diagnosed if the patient met the following criteria: (a) meets the core clinical criteria for MCI, and (b) has positive biomarkers for both Aβ and neuronal injury. For the diagnosis of PD-MCI, we used the Movement Disorder Society Task Force diagnostic criteria and classified patients with either clinically established Parkinson’s disease or clinically probable Parkinson’s disease [11]. The diagnosis of PD-MCI was made if participants met the following criteria: (a) meets the core clinical criteria for MCI, and (b) diagnosis of Parkinson’s disease as based on the UK PD Brain Bank Criteria [16]. Patients with MCI due to cerebrovascular disease, other etiologies, or at least two etiologies were excluded.

### 2.3. Assessment of Fear of Falling

We used the Activities-specific Balance Confidence (ABC) scale to measure FOF. The ABC scale is a 16-item self-report measure of balance confidence in performing activities without losing balance [17]. This scale was shown to have good validity [18] and excellent test-retest reliability [17] in measuring FOF in community-dwelling older adults. The participants are asked to rate their confidence in performing various activities without losing their balance or becoming unsteady, where 0 is “no confidence” and 100 is “completely confident”. The overall score is calculated by adding the item scores and dividing the total by 16. 

### 2.4. Outcome Measures

Participants included in the study underwent comprehensive neuropsychological testing and physical activity testing.

The neuropsychological assessment included the Mini-Mental State Examination, parts A and B of the Trail Making Test, the category fluency test, the forward and backward digit recall tests, the California Verbal Learning Test, the Judgment of Line Orientation test-Short Form, the Boston Naming Test, the Geriatric Depression Scale-15, and the General Anxiety Disorder-7 questionnaire. The Mini-Mental State Examination is a widely used and valid test of global cognitive function among elderly individuals [19]. The parts A and B of the Trail Making Test was designed to measure visual attention and executive function [20]. The category fluency test is one of the most commonly used measures of executive function and language [21]. The forward and backward digit recall tests are neuropsychological tests of short-term verbal language and executive function [22]. The California Verbal Learning Test is a comprehensive, detailed assessment of episodic memory in older adults [23]. The short form of the Judgment of Line Orientation test is a commonly used measure of visuospatial perception [24]. The Boston Naming Test is a valid assessment tool for the measure of confrontation naming in individuals with language impairments caused by stroke, Alzheimer’s disease and dementia [25]. The Geriatric Depression Scale-15 is a widely used assessment tool to evaluate the prevalence of depressive symptoms in the older adults [26]. The General Anxiety Disorder-7 questionnaire was designed to screen for anxiety or to measure its severity [27].

Physical activity was assessed using the Tinetti assessment scale, straight walking performance, and the timed up and go test. The Tinetti assessment scale is a very good indicator of the fall risk of an older adult and evaluates balance ability and gait performance [28]. The G-WALK^®^ (BTS Bioengineering Corp., Quincy, MA, USA) system, which is a wearable system for the functional analysis of movement, was used for straight walking performance and the TUG test. This system comprises a portable inertial sensor placed in the area of the S1–S2 vertebrae to record specific movements. The validity and reliability of this system to evaluate movement performance has been well established [29]. In the straight walking test, the participants were instructed to walk straight for 10 m at a usual speed three times. In the TUG test, the participants were asked to rise from a 45 cm-high chair, walk 3 m, turn around, return, and sit down three times. The total TUG duration was divided into four subtasks: time to stand (sit-to-stand), time to turn around in the midway (turn-to-walk), time to turn around to reach the chair (turn-to-sit), and time to sit down in the chair (stand-to-sit). The average of the three trials in each test was used for data analysis.

### 2.5. Statistics

Statistical analyses were conducted using SPSS 22.0 (SPSS, Inc., Chicago, IL, USA). Descriptive data were reported in terms of means, SDs or numbers. Independent t-tests (continuous variables) or chi-square tests (categorical variables) were used to compare the between-group differences. In this study, the correlations were first established, and the factors significantly correlating with ABC scores were further analyzed using multivariable linear regression models. Covariates in multivariable models included age, sex, fall history, education, and comorbidities (e.g., hypertension, diabetes mellitus and cardiovascular disease). Spearman’s rank correlation analysis was used to examine correlations between ABC scores and cognitive performance and gait performance. The level of significance was set at a *p* value of less than 0.05.

## 3. Results

### 3.1. Baseline Demographic Data and Comparisons between AD-MCI and PD-MCI

Seventy-one participants (male: 38; female: 33) participated in this study. Table 1 presents the characteristics and performance of the participants in the AD-MCI (N = 37) and PD-MCI (N = 34) groups. The mean ages of the participants in the AD-MCI and PD-MCI groups were 72.4 ± 8.8 and 68.4 ± 10.2 years, respectively. There were no statistically significant differences in the basic characteristics (sex, age, education, onset duration, body mass index, functional status and falls history), cognitive functions, depression, anxiety, straight walking performance, or performance in two timed up and go subtasks (sit-to-stand and stand-to-sit) between the two groups. However, the participants with PD-MCI had significantly lower ABC scale scores (*p* = 0.002), which indicated less confidence in performing activities without losing balance, than did the AD-MCI participants. In addition, compared to the patients with AD-MCI, those with PD-MCI had significantly lower Tinetti gait and balance scale scores (Tinetti gait, *p* = 0.018; Tinetti balance, *p* = 0.019) and longer completion times on the two timed up and go turning subtasks (turn-to-walk, *p* = 0.007; turn-to-sit, *p* = 0.022).

### 3.2. Factors Determining FOF

The correlation coefficients between cognitive performance and ABC scores are shown in Table 2. In the AD-MCI group, the ABC scores were positively correlated with Mini-Mental State Examination, forward digits recall test and backward digits recall test scores (r = 0.414 to 0.516, *p* < 0.05), meaning that greater balance confidence was associated with better global cognition, attention, and working memory performance. The Trail Making Test B scores were negatively correlated with the ABC scores (r = −0.409, *p* < 0.05), meaning that less balance confidence was associated with worse visual attention and executive function. In the PD-MCI group, there were no significant differences between cognitive functions and ABC scores.

The correlation coefficients between balance and gait performance and ABC scores are shown in Table 3. The gait speed, stride length, cadence, and Tinetti assessment scale scores were positively correlated with the ABC scores in the AD-MCI group (r = 0.336 to 0.507, *p* < 0.05), meaning that greater balance confidence was associated with better balance and gait performance. In the PD-MCI group, gait speed was positively correlated with ABC scores (r = 0.385, *p* < 0.05), meaning that greater balance confidence was associated with better gait speed. Performance on the timed up and go subtasks was negatively correlated with the ABC scores (r = −0.503 to −0.663), *p* < 0.05), meaning that less balance confidence was associated with worse functional mobility.

Based on the regression models (Table 4), after adjusted for age, sex, fall history, education, and comorbidities, attention and working memory were the most important factors in determining ABC scores in the AD-MCI group (F = 10.162, effect size f2 = 0.328, power = 0.92, *p* = 0.003). On the other hand, turn-to-walk time was the most important factor in determining ABC scores in the PD-MCI group (F = 35.292, effect size f2 = 1.179, power = 0.99, *p* < 0.001).

## 4. Discussion

This study investigated the factors associated with FOF in older people with different types of MCI (i.e., AD-MCI and PD-MCI). The results indicated that attention and working memory were the most important factors in determining ABC scores in the AD-MCI group, whereas turn-to-walk was the most important factor in determining ABC scores in the PD-MCI group.

A previous study indicated that older adults with cognitive impairments have greater FOF [30], and the prevalence of FOF in those with MCI was significantly higher than that in cognitively healthy older adults. Interestingly, we found different results in the two MCI groups. In the PD-MCI group, the ABC scale score was significantly lower than that in the AD-MCI group, which indicated that the individuals with PD-MCI had less confidence/greater FOF in performing activities than those with AD-MCI. This might be due to poor balance and gait performance in the PD-MCI group. However, the ABC scale score in the AD-MCI group was not only significantly higher than that in the PD-MCI group but also significantly higher than that in age-matched controls (90.28 vs. 79.89 from Huang et al. [31], one-sample t-test, *p* < 0.001). This result indicated that the participants with AD-MCI had more confidence/less FOF in conducting activities of daily living that required transferring, bending, reaching, or walking than cognitively healthy elderly participants. However, the prevalence of falls in older individuals with MCI and Alzheimer’s disease is higher than that in cognitively healthy individuals [2]. Do they truly have confidence in engaging in daily activities, or do they lack an awareness of the risk of falling? Henry et al. found that patients with mild dementia had difficulties differentiating high- from low-danger situations, and this difficulty was related to more general cognitive decline [32]. A cross-sectional study reported that cognitive impairment was significantly associated with the absence of FOF in community-living older individuals, which indicated that worse cognitive function may inhibit FOF [33]. Thus, the participants with AD-MCI in the present study may have impaired threat perception, which led to lower FOF. Lack of FOF might lead to subsequent fall incidents [34]. Future studies can further investigate this interesting and important point.

In the AD-MCI group, global cognitive function, executive function, attention and working memory were significantly correlated with ABC scores. The regression model showed that attention and working memory were the most important factors in determining FOF. There have been inconsistent findings on the relationship between FOF and cognition in the aging population. Shirooka et al. indicated that cognitive impairment, especially executive function, is related with the absence of FOF in older adults with frailty [33]. A one-year longitudinal study of 406 community-dwelling older adults reported that the presence of subjective memory complaints at baseline was related to FOF [35]. Another longitudinal study of 4931 middle-age and older adults showed that FOF was associated with a greater odds of decreased Montreal Cognitive Assessment but not Mini-Mental State Examination scores [36]. A 3 year prospective study of 4280 older adults with normal cognition noted that the older adults who were very afraid of falling were 1.45 times more likely to cognitive decline than those who were not afraid of falling [37]. These inconsistent results demonstrated that cognitive impairment may play different roles in determining FOF in different aging populations. In our study, we found that the most important factors in determining FOF in AD-MCI, but not PD-MCI, were attention and working memory. Previous study indicated that the cognitive worsening over time is associated with a worse motor performance at baseline [38]. Attention and working memory are crucial for postural and gait control, and adding a concurrent secondary task while walking or maintaining balance requires even more attention, especially in older adults [39]. Impaired attention and working memory may have a negative effect on postural control, which leads to FOF. The finding of present study appears in line with a broad psychogeriatric perspective, regarding the peculiar involvement of psychological factors along cognitive-physical trajectories of aging. 

In the PD-MCI group, the turn-to-walk measure was the most important factor in determining ABC scores. A cross-sectional study of 104 people with Parkinson’s disease reported that the strongest contributing factor to FOF was walking difficulties [40]. Haertner et al. indicated that FOF was associated with a longer turning duration in individuals with Parkinson’s disease, which is in line with our result [41]. A novel finding in this study was that even in the presence of cognitive degradation, turning performance remained the most important factor associated with FOF in individuals with Parkinson’s disease. Parkinson’s disease is a neurodegenerative disease that leads to motor dysfunctions, including walking difficulties. With the progression of the disease, people with Parkinson’s disease show impairments in cognitive functions. Impaired cognitive function has been suggested to be related to FOF [42]. However, our regression models indicated that turning performance, not cognitive function, was the most important factor contributing to FOF in persons with Parkinson’s disease. Turning is a complex movement that requires interlimb coordination, posture and gait control, and continuous movement of the center of gravity [43]. Turning difficulties are a common problem in Parkinson’s disease patients due to impairments in balance control and freezing of gait that can lead to falls [44]. When compared with healthy subjects, people with Parkinson’s disease need more time to turn, have a narrower base of support, and usually present freezing of gait during turning periods [45]. These disturbances not only influence the quality of their turning performance but can also cause FOF.

FOF has been shown to be a major barrier to performing daily activities and to engaging in exercise, which may cause social isolation and functional decline in the older adults. Thus, there is a need to understand the factors contributing to the FOF to efficiently address this in clinical practice and research. Based on our results, attention and working memory were the most important factors in determining FOF in the AD-MCI group, and the turn-to-walk ability was the most important factor in determining FOF in the PD-MCI group. We suggest that training programs that focus on improving cognitive function and turning performance may decrease FOF in individuals with AD-MCI and PD-MCI, respectively. Parry et al. reported that cognitive-behavioural therapy probably reduces FOF and depression in community-living older adults with undue FOF [46]. A meta-analysis of thirty trials noted that exercise intervention may have a small to moderate positive effect immediately after intervention in community-dwelling older individuals [47]. Therefore, future studies could design new cognitive and exercise programs based on our results and explore their effects on FOF in aging population. 

There were several limitations should be pointed out in this study. Firstly, the relatively small sample size may limit the strength of the results. Future studies should have a larger sample size to confirm our results. Moreover, we used a cross-sectional study design, which limited our exploration of the predictive factors of FOF in these two groups. To better understand the cause-effect relationship between contributing factors and FOF in older adults with high risk of cognitive impairment, a large cohort study is encouraged. Furthermore, we did not measure a parameter for the severity of motor impairment such as Unified Parkinson’s Disease Rating Scale (MDS-UPDRS) score in PD-MCI group. The turn-to-walk parameter was the most important parameter explaining FOF, which may be explained by motor impairment. Treatment effects of cognitive training and turning performance training to improve FOF in MCI populations are warranted and encouraged for future studies.

## 5. Conclusions

The present study found that FOF in individuals with different types of MCI was influenced by different factors. Attention and working memory were the most important contributing factors in the AD-MCI group, whereas the turn-to-walk ability was the most important contributing factor in the PD-MCI group. Thus, therapies that aim to lower FOF in AD-MCI and PD-MCI populations may address attention and working memory and turn-to-walk performance, respectively.

## Figures and Tables

**Figure 1 brainsci-12-00990-f001:**
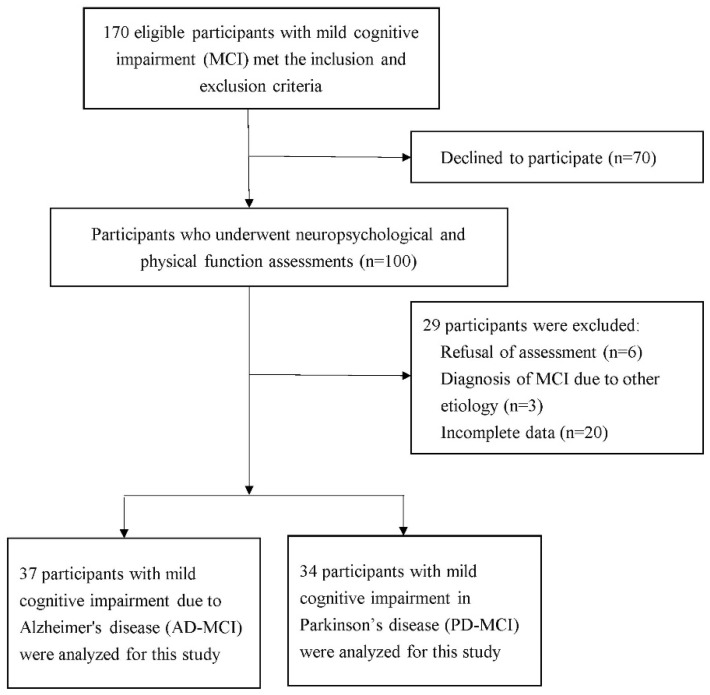
Flow chart showing the process of selecting subjects in this study.

**Table 1 brainsci-12-00990-t001:** Comparison of participants’ characteristics between the AD-MCI and PD-MCI groups (*n* = 71).

Variable	Total (*n* = 71)	AD-MCI (N = 37)	PD-MCI (N = 34)	*p*-Value
Gender (male/female)	38/33	20/17	18/16	0.936
Age (years)	70.46 ± 9.64	72.41 ± 8.84	68.35 ± 10.16	0.077
Education (years)	7.85 ± 3.95	7.03 ± 4.11	8.74 ± 3.61	0.068
Onset duration (years)	1.83 ± 1.80	2.11 ± 1.91	1.53 ± 1.54	0.167
Falls history (*n*)	28	11	17	0.081
BMI	23.93 ± 3.42	23.86 ± 3.6	24.00 ± 3.27	0.861
Barthel Index	98.21 ± 7.42	99.59 ± 1.82	96.67 ± 10.51	0.124
IADL	21.54 ± 4.76	22.27 ± 4.37	20.73 ± 5.11	0.178
Hypertension (*n*)	42	26	16	0.047
Diabetes mellitus (*n*)	20	12	8	0.405
Cardiovascular disease (*n*)	25	19	6	0.003
Global cognitive function				
MMSE	25.32 ± 3.57	24.65 ± 4.10	26.06 ± 2.79	0.096
Episodic memory				
CVLT-SF	18.79 ± 5.40	18.70 ± 5.36	18.88 ± 5.51	0.890
Visuospatial performance				
Judgment of line orientation	13.51 ± 3.56	13.25 ± 3.23	13.79 ± 3.91	0.534
Attention and working memory				
Forward digits	7.24 ± 1.53	7.43 ± 1.56	7.03 ± 1.49	0.269
Backward digits	3.97 ± 1.52	4.03 ± 1.67	3.91 ± 1.38	0.753
Executive function				
Category fluency test	12.45 ± 3.79	12.57 ± 3.87	12.32 ± 3.76	0.789
Trail making test A (s)	25.60 ± 15.57	24.51 ± 15.22	26.81 ± 16.09	0.542
Trail making test B (s)	67.00 ± 36.49	63.67 ± 36.67	70.65 ± 36.50	0.431
Language				
Boston naming test	22.39 ± 5.24	21.78 ± 5.48	23.06 ± 4.97	0.309
Depression				
GDS-15	3.07 ± 3.34	2.76 ± 3.11	3.42 ± 3.60	0.408
Anxiety				
GAD-7	2.43 ± 3.67	2.00 ± 3.66	2.91 ± 3.68	0.304
Fear of falling				
ABC scale	82.40 ± 22.60	90.28 ± 16.10	73.58 ± 25.62	0.002
Balance and gait performance				
Tinetti gait	14.91 ± 2.52	15.62 ± 1.21	14.12 ± 3.30	0.018
Tinetti balance	11.44 ± 1.52	11.86 ± 0.67	10.97 ± 2.01	0.019
Straight walking performance				
Gait speed (m/s)	0.84 ± 0.22	0.87 ± 0.22	0.82 ± 0.21	0.302
Stride length (cm)	63.01 ± 19.39	63.36 ± 17.38	62.63 ± 21.57	0.876
Cadence (steps/min)	97.86 ± 14.18	97.60 ± 11.39	98.13 ± 16.81	0.878
TUG subtasks				
Sit-to-stand (s)	1.83 ± 0.57	1.74 ± 0.58	1.91 ± 0.55	0.209
Turn-to-walk (s)	2.65 ± 1.34	2.23 ± 0.80	3.11 ± 1.64	0.007
Turn-to-sit (s)	2.50 ± 1.53	2.09 ± 0.93	2.96 ± 1.91	0.022
Stand-to-sit (s)	2.45 ± 0.89	2.40 ± 0.83	2.51 ± 0.97	0.593

Abbreviation: AD-MCI, mild cognitive impairment due to Alzheimer’s disease. PD-MCI, mild cognitive impairment in Parkinson’s disease. BMI, body mass index. MMSE, Mini-Mental State Examination. IADL, Instrumental Activities of Daily Living. GDS-15, Geriatric Depression Scale-15. GAD-7, General Anxiety Disorder-7 questionnaire. ABC, Activities-specific Balance Confidence Scale.

**Table 2 brainsci-12-00990-t002:** Correlation coefficients between cognitive performance and ABC scores.

Group		AD-MCI Group (N = 37)		PD-MCI Group (N = 34)	
		ABC Score		ABC Score	
Domain	Outcome Measures	r	*p*-Value	r	*p*-Value
Cognition	MMSE	0.431 *	0.008	0.244	0.165
	CVLT-SF	0.315	0.058	0.221	0.209
	Judgment of line orientation	0.217	0.203	0.011	0.952
	Forward digits	0.516 *	0.001	0.297	0.088
	Backward digits	0.414 *	0.012	0.117	0.510
	Category fluency test	0.155	0.360	0.279	0.109
	Trail making test A	−0.290	0.082	−0.319	0.070
	Trail making test B	−0.409 *	0.013	−0.221	0.217
	Boston naming test	0.234	0.164	0.252	0.150
Depression	GDS-15	−0.250	0.136	−0.056	0.754
Anxiety	GAD-7	−0.218	0.195	−0.173	0.329

Abbreviation: AD-MCI, mild cognitive impairment due to Alzheimer’s disease. PD-MCI, mild cognitive impairment in Parkinson’s disease. MMSE, Mini-Mental State Examination. CVLT-SF, California verbal language test-short form. GAD-7, General Anxiety Disorder-7 questionnaire. GDS-15, Geriatric Depression Scale-15. ABC, Activities-specific Balance Confidence Scale. * *p* < 0.05.

**Table 3 brainsci-12-00990-t003:** Correlation coefficients between gait performance and ABC scores.

Group		AD-MCI Group (N = 37)		PD-MCI Group (N = 34)	
		ABC Score		ABC Score	
Domain	Outcome Measures	r	*p*-Value	r	*p*-Value
Gait performance	Speed (m/s)	0.507 *	0.002	0.385 *	0.027
	Stride Length	0.439 *	0.007	0.321	0.065
	Cadence step	0.349 *	0.037	0.119	0.503
Tinetti assessment scale	Tinetti Posture	0.482 *	0.003	0.165	0.352
	Tinetti Gait	0.336 *	0.042	0.317	0.068
TUG subtasks	Turn-to-walk	−0.316	0.056	−0.663 **	<0.001
	Turn-to-sit	−0.302	0.069	−0.464 *	0.006
	Sit-to-stand	−0.292	0.079	−0.503 *	0.002
	Stand-to-sit	−0.138	0.415	−0.229	0.192

Abbreviation: AD-MCI, mild cognitive impairment due to Alzheimer’s disease. PD-MCI, mild cognitive impairment in Parkinson’s disease. ABC, Activities-specific Balance Confidence Scale. TUG, timed up and go test. * *p* < 0.05; ** *p* < 0.005.

**Table 4 brainsci-12-00990-t004:** Summary of linear regression analyses in the AD-MCI and PD-MCI groups.

AD-MCI Group			PD-MCI Group		
Variable	β	Coefficient	Variable	β	Coefficient
Age	−0.338	−0.344	Age	−0.113	−0.015
Sex	−0.198	−0.224	Sex	−0.089	−0.132
Fall history	−0.056	−0.063	Fall history	−0.191	−0.277
Education level	0.168	0.192	Education level	0.113	0.167
Hypertension	−0.037	−0.042	Hypertension	0.019	0.028
Diabetes mellitus	0.001	0.001	Diabetes mellitus	0.197	0.291
Cardiovascular disease	−0.286	−0.328	Cardiovascular disease	−0.020	−0.029
MMSE	0.176	0.188	Speed	0.068	0.080
Forward digits	0.497		Turn-to-walk	−0.735	
Backward digits	0.049	0.047	Turn-to-sit	−0.021	−0.014
Trail making test B	−0.115	−0.127	Sit-to-stand	−0.136	−0.173
Speed	0.268	0.297	R square	0.541	
Stride Length	0.389	0.443	Adjusted R square	0.525	
Cadence step	−0.047	−0.052	*p* value	<0.001	
Tinetti Posture	0.071	0.081			
R square	0.247				
Adjusted R square	0.223				
*p* value	0.003				

Abbreviations: AD-MCI, mild cognitive impairment due to Alzheimer’s disease. PD-MCI, mild cognitive impairment in Parkinson’s disease. MMSE, Mini-Mental State Examination. Models: adjusted for age, sex, fall history, education, and comorbidities (e.g., hypertension, diabetes mellitus and cardiovascular disease).

## Data Availability

The datasets used and/or analysed during the current study are available from the corresponding author on reasonable request.

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
