# Peer review of "Factors Associated with Fear of Falling in Individuals with Different Types of Mild Cognitive Impairment"

_brainsci, 2022, doi:10.3390/brainsci12080990_

Round 1

Reviewer 1 Report

I found the study interesting, and the findings stimulating. 

Please, find few comments below, in order to improve  methodological and theoretical contents.

Manuscript ID: brainsci-1826248
Title: Factors Associated with Fear of Falling in Individuals with Dif-2 ferent Types of Mild Cognitive Impairment
Comments
a. Line 42: Within a geriatric perspective, “disability” and “functional dependence” are considered close to synonymies, since both refers to a reduction of autonomy in performing daily activities. Did the Authors want to underline a peculiar difference between the two terms?
b. Line 43: The term “psychological difficulties” appears too vague. Certainly, fear of falling should not be considered as a one of “psychological difficulties”. I would rephrase this part, as follows: “Older adults, especially those at risk of falls, may also exhibit peculiar psychological features, such as fear of falls, linked to the experience of negative emotions related to falls”.
c. Materials and Methods: Was the presence of positive psychiatric history considered as putative exclusion criteria? If not, it should be mentioned as a limitation of the study.
d. Table 4 should be edited, in order to better and clearly describe the regression model, in each step, including each relevant coefficient.
e. The findings of the study appears in line with a broad psychogeriatric perspective, regarding the peculiar involvement of psychological factors along cognitive-physical trajectories of aging. In my opinion, this implication should be reinforced in the Discussion. I would merely suggest a recent contribution, which the Authors may refer to.
Quattropani MC, Sardella A, Morgante F, et al. Impact of Cognitive Reserve and Premorbid IQ on Cognitive and Functional Status in Older Outpatients. Brain Sci. 2021;11(7):824. Published 2021 Jun 22. doi:10.3390/brainsci11070824

Author Response

Dear reviewer,

Thank you for the constructive and positive review of our manuscript entitled Factors Associated with Fear of Falling in Individuals with Different Types of Mild Cognitive Impairment. In the following, we have addressed all the comments on a point-by-point basis. We hope that the revisions are satisfactory.

1. Line 42: Within a geriatric perspective, “disability” and “functional dependence” are considered close to synonymies, since both refers to a reduction of autonomy in performing daily activities. Did the Authors want to underline a peculiar difference between the two terms?

Response: We thank you for the comment. We agree with the reviewer’s comment that “disability” and “functional dependence” are considered close to synonymies. Thus, we have deleted the redundant words “functional dependence” in the revised manuscript. Please see page 1, line 42.

2. Line 43: The term “psychological difficulties” appears too vague. Certainly, fear of falling should not be considered as a one of “psychological difficulties”. I would rephrase this part, as follows: “Older adults, especially those at risk of falls, may also exhibit peculiar psychological features, such as fear of falls, linked to the experience of negative emotions related to falls”.

Response: We thank you for the suggestion. The revised manuscript has been rephrased. Please see page 1, lines 42-43.

3. Materials and Methods: Was the presence of positive psychiatric history considered as putative exclusion criteria? If not, it should be mentioned as a limitation of the study.

Response: We thank you for the comment. We did not recruit any participants who has any psychiatric history. Thus, we have added this exclusion criteria in the revised manuscript. Please see page 2, lines 90-91.

4. Table 4 should be edited, in order to better and clearly describe the regression model, in each step, including each relevant coefficient.

Response: We thank you for the suggestion. We have reedited the Table 4 in the revised manuscript. Please see page 7, Table 4.

5. The findings of the study appears in line with a broad psychogeriatric perspective, regarding the peculiar involvement of psychological factors along cognitive-physical trajectories of aging. In my opinion, this implication should be reinforced in the Discussion. I would merely suggest a recent contribution, which the Authors may refer to.

Quattropani MC, Sardella A, Morgante F, et al. Impact of Cognitive Reserve and Premorbid IQ on Cognitive and Functional Status in Older Outpatients. Brain Sci. 2021;11(7):824. Published 2021 Jun 22. doi:10.3390/brainsci11070824

Response: We thank you for the suggestion. We have added this discussion and also have cited the reference in the revised manuscript. Please see page 8, lines 276-277.

Reviewer 2 Report

The study is well-planned and explained in terms of both novelty and experimental procedure. Though authors need to discuss more about the relevance of this data in developing the therapies for the treatment of FOF in AD-MCI and PD-MCI populations.

Also, a recent study by Mayuka Oki et al., 2021, claimed that cognitive impairment is not a risk factor in FOF. Authors should discuss such contradict studies in their discussion to provide a good validation for their own study.

Author Response

Dear reviewer,

Thank you for the constructive and positive review of our manuscript entitled Factors Associated with Fear of Falling in Individuals with Different Types of Mild Cognitive Impairment. In the following, we have addressed all the comments on a point-by-point basis. We hope that the revisions are satisfactory.

1. The study is well-planned and explained in terms of both novelty and experimental procedure. Though authors need to discuss more about the relevance of this data in developing the therapies for the treatment of FOF in AD-MCI and PD-MCI populations.

Response: We thank you for the suggestion. We have added this discussion in the revised manuscript. Please see page 9, lines 310-317.

2. Also, a recent study by Mayuka Oki et al., 2021, claimed that cognitive impairment is not a risk factor in FOF. Authors should discuss such contradict studies in their discussion to provide a good validation for their own study.

Response: We thank you for the suggestion. However, we only found one article published by Mayuka Oki et al., 2021 and this article was exploring risk factors for falls, not fear of falling. Thus, we did not discuss this study in the revised manuscript.   

Round 2

Reviewer 1 Report

Dear Authors, thank you for having welcomed my suggestions.

I found the manuscript improved in its overall quality.